# Can You Follow Me?
# Testing Situational Understanding in ChatGPT

**Chenghao Yang**
University of Chicago
chenghao@uchicago.edu

**Allyson Ettinger**
Allen Institute for AI
allysone@allenai.org

## Abstract

Understanding sentence meanings and updating information states appropriately across time—what we call "situational understanding" (SU)—is a critical ability for human-like AI agents. SU is essential in particular for chat models, such as ChatGPT, to enable consistent, coherent, and effective dialogue between humans and AI. Previous works have identified certain SU limitations in non-chatbot Large Language models (LLMs), but the extent and causes of these limitations are not well understood, and capabilities of current chat-based models in this domain have not been explored. In this work we tackle these questions, proposing a novel synthetic environment for SU testing which allows us to do controlled and systematic testing of SU in chat-oriented models, through assessment of models' ability to track and enumerate environment states. Our environment also allows for close analysis of dynamics of model performance, to better understand underlying causes for performance patterns. We apply our test to ChatGPT, the state-of-the-art chatbot, and find that despite the fundamental simplicity of the task, the model's performance reflects an inability to retain correct environment states across time. Our follow-up analyses suggest that performance degradation is largely because ChatGPT has non-persistent in-context memory (although it can access the full dialogue history) and it is susceptible to hallucinated updates—including updates that artificially inflate accuracies. Our findings suggest overall that ChatGPT is not currently equipped for robust tracking of situation states, and that trust in the impressive dialogue performance of ChatGPT comes with risks. We release the codebase for reproducing our test environment, as well as all prompts and API responses from ChatGPT, at https://github.com/yangalan123/SituationalTesting.

## 1 Introduction

Understanding the meaning of language inputs and their impact on information states is essential for building a communicative human-like AI agent (e.g., ChatGPT (OpenAI, 2022)). This capability requires an agent to know **truth conditions** (Lewis, 1976; Heim and Kratzer, 1998) of a given input, and how that input **updates the context** over time (Veltman, 1996). For instance, the natural language input "Open BOX-4, obtain KEY-1" in a game environment should assert updates "OPENED(BOX-4) = True" and "OBTAINED(KEY-1) = True" in the agent's representation of the environment state. We describe this ability as situational understanding (SU), as the agent needs to ground language in situations and understand **situational changes**.

Recent research indicates that despite the tremendous success of Large Language Models (LLMs) (e.g., GPT-3 (Brown et al., 2020)), these models still fail to understand situational changes, and cannot serve as human-like agents to accomplish real-world tasks. For example, evidence suggests that models fail to detect sarcasm expressed in underlying context (Suzgun et al., 2022), infer incorrect logic states when context changes in games (Li et al., 2022) and fail to track entity state changes in a discourse (Kim and Schuster, 2023). While these works have shown important limitations in LLMs, it remains unclear why these models show these limitations, and there is less work that shows the extent to which these limitations persist in more recent, chat-trained models like ChatGPT.

In this work we seek to shed light on both of these questions. To test the situation tracking ability of ChatGPT and other chat models, we design a synthetic environment for controlled testing of models' ability to follow instructions and maintain consistent understanding of situation states. We design a multi-stage testing framework, and include various controls to avoid models relying on mappings memorized from pre-training rather than doing real-time state tracking. Our environment allows for flexible testing under different conditions,

as well as controlled follow-up analyses to enable closer examination of causal factors.

We apply our tests primarily to ChatGPT, which we find to outperform other models fine-tuned on chats. Our results show that ChatGPT's performance reflects a failure to retain coherent and correct environment states across time, despite the simplicity of the task and the fact that ChatGPT has access to the full dialogue history in its input window. Our follow-up analyses suggest that a major contributor to this degradation is failure to retain prior states in memory, as well as susceptibility to spurious hallucinated updates (which can also contribute to accidentally correct outputs). Our findings suggest overall that ChatGPT lacks the ability to track situation state changes robustly.

## 2 Related work

Situational understanding ability is essential for building agent-like real-world intelligent systems. In comprehensive benchmarks, Big-Bench-Hard (BBH, Suzgun et al. (2022)) finds that for GPT-family models and PaLM 540B models (Chowdhery et al., 2022), even equipped with the state-of-the-art Chain-of-Thought prompting, still fail on tasks that require situational information (e.g., to detect sarcasm). HELM (Liang et al., 2022) also points out LLMs can lack state tracking ability based on evaluation results for 30 models on bAbI (Weston et al., 2016) environment.

In synthetic and controlled environments, Li et al. (2021); Andreas (2022); Li et al. (2022) does probing on BART (Lewis et al., 2020), T5 (Raffel et al., 2020) and GPT-3 using the SCONE (Long et al., 2016) and TextGame (Côté et al., 2019) datasets, and show that these models lack the ability to track and infer information states. Kim and Schuster (2023) re-analyze the results in Li et al. (2021) and find that GPT-3/3.5 and Flan-T5 (Chung et al., 2022) cannot track entity state changes in a discourse. Toshniwal et al. (2022) instruct GPT-2 (Radford et al., 2019) to play chess and find it difficult to track board states per move.

We build on these previous works in two primary ways. First, these works mainly use LLMs as feature extractors (e.g., train probe models over intermediate representations or final-layer representations), and none of them discuss whether situational understanding limitations still exist in the recent powerful and widely used ChatGPT system. We investigate here whether ChatGPT has the crit-

ical underlying situational understanding to generate coherent, consistent and effective dialogue (Karttunen, 1969; Kamp et al., 2011; Wu et al., 2019). Second, tests used in previous work are susceptible to interference from confounds present when fine-tuning linear probes, or shortcuts in the testing environment that can be utilized to bypass situational understanding tests (Kim and Schuster, 2023). We aim to address some of these concerns by introducing a number of additional controls in our tests. Finally, we also carry out a more in-depth exploration of model update dynamics, to better understand causes for patterns of performance.

## 3 Building a Situational Testing Environment

What we want to test is models' ability to process situational changes conveyed by language inputs, and to maintain internal representations of the corresponding situation states. To test this, we use a synthetic box-moving environment like TextGame (Li et al., 2021), where we have full access to underlying game states, but eliminate complicated reward pursuing and route branching. Having this kind of full-information synthetic environment is helpful to test models' evolving understanding of environment states as the input describes progressively more situational changes.

### 3.1 Environment Setup

The environment includes two basic components:

1. **Instructions.** Instructions directed to an agent, defining the agent's quest and providing information about the environment. As we will describe below, this component also sometimes contains explanations of the meanings of non-language functors and arguments to be used in the output.

2. **Steps and queries.** Descriptions of steps taken by the agent, followed by a query that prompts for enumeration of all relevant environment states after the action is taken. Queries and answers to the queries take the format of sets of logical representations of states with corresponding truth values.

In our tests, the input provided to the model includes the task instructions, along with $n$ few-shot Step-Query-Answer examples to demonstrate the task. In the zero-th step, we define the step

as "Do nothing" and provide in the Answer the full set of correct truth values as an initialization. The $n$ few-shot Step-Query-Answer examples are then followed by a series of Steps without Query-Answer components, followed by a Test Step and Test Query that the model is expected to complete with the full enumeration of environment states that hold after that step is taken. An example of our environment format, to be further explained below, is shown in Example 1.

```
1  Instructions: As an agent, you need to find
       the way to go out of this quest.
       Currently, there are several boxes in
       front of you and there is a key inside
       each box. You can use only one of these
        keys to open the door and finish this
       quest. There are 5 boxes and 5 keys
       here. Boxes are identified as jqC-X and
        Keys are identified as bsS-X.
       NvSWxzvJb(jqC-2)=True means that jqC-2
       has been opened. B(bsS-3)=True means
       that bsS-3 has been obtained. NvSWxzvJb
       (jqC-2)=False means that jqC-2 has not
       been opened. B(bsS-3)=False means that
       bsS-3 has not been obtained.
2  Step-0: Initialization. Do nothing.
3  Question: NvSWxzvJb(jqC-0)=?...B(bsS-0)=?B(
       bsS-1)=?...
4  Answer: NvSWxzvJb(jqC-0)=False...B(bsS-0)=
       False...
5  Step-1: Open jqC-3 and retrieve bsS-2.
6  Question: NvSWxzvJb(jqC-0)=?...
```

Example 1: Running example of our test environment.

For the experiments below, we set the number of boxes and keys to 10. This corresponds to a total of 20 environment states (boxes and keys have separate states) to be enumerated after each step. We sample the number of steps randomly from a uniform distribution $U(1, 10)$. We then keep instructions almost entirely the same across samples, except that in the Synthetic Language settings (see Section 3.2) the state predicates are defined randomly and therefore vary between samples (e.g., in one instance we use *NvSWxzvJb(jqC-0)=True* to refer to the box *jqC-0* having been opened, while in another instance we use *Abc(bb-0)=True* to represent the box *bb-0* having been opened).

## 3.2 Robustness Checks for State Tracking

**Synthetic language**   When testing models' ability to map to logical representations of environment states, a concern with using language-based logical symbols (such as OPENED(BOX-4)) is that the models may be able to leverage pre-training on similar language-based logical symbols, or simply copy from the input language describing the

actions, without needing to convert to abstract situation states. To control for this possibility, in addition to using natural language (NL) functors and arguments for our environment states, we also adopt settings in which **synthetic language** (SL) is used to specify the functors and arguments of the environment states. This allows us to disentangle our target task from pattern memorization and copying capabilities, better ensuring that models must rely on a combination of the instructions and the changes caused by the actions taken. To build synthetic functors and arguments, we use randomly selected ASCII characters. The length for each functor or argument is a random sample from $U(1, 10)$. In the instructions, we include explanations of the meanings of these synthetic functors and arguments. An example of our synthetic language setting is shown in Example 1 (example instructions from our NL setting can be seen in Example 2 and Example 3). To counteract randomness effects and reduce any bias from specific synthetic language, for each test case (an **instance**), we generate a different set of synthetic functors and arguments.

**Counterintuitive instructions**   To further control for potential memorization of mappings between language and logical states from pre-training, we include one additional manipulation involving what we call **counterintuitive instruction**. In counterintuitive instruction settings, the instructions define mappings that reverse the standard usage of logical statements and truth values. The two versions we use are *counterintuitive output format* (Example 2), and *counterintuitive language instruction* (Example 3). These manipulations draw on the tradition of negation testing (McCoy et al., 2019; Ribeiro et al., 2020; Hossain et al., 2020; Ravichander et al., 2022), and allow us to further disentangle pre-training memorization from understanding of our particular instructions.[1]

## 3.3 Evaluation Metrics

We mainly evaluate models' success in our synthetic environment by measuring two metrics: **State-EM**, and **Step-EM**.

**State-EM** is the proportion of all predicted

---

[1]There are other possible perturbations that we can do, such as adding a distractor at each step (Appendix A) or applying synthetic language only on functors / only on arguments (Appendix B). For simplicity, we omit the discussion of other perturbation types and only focus on the two robustness checks explained in this section.

```
7  Instructions: ... OPENED(BOX-3)=False
      means that BOX-3 has been opened.
      OBTAINED(KEY-1)=False means that KEY
      -1 has been obtained. OPENED(BOX-3)=
      True means that BOX-3 has not been
      opened. OBTAINED(KEY-1)=True means
      that KEY-1 has not been obtained.
```

Example 2: CounterIntuitive Output Format

```
8  Instructions: ... OPENED(BOX-3)=True
      means that BOX-3 has Not been opened
      . OBTAINED(KEY-1)=True means that
      KEY-1 has Not been obtained. OPENED(
      BOX-3)=False means that BOX-3 has
      not been opened. OBTAINED(KEY-1)=
      False means that KEY-1 has not been
      obtained.
```

Example 3: Counter-Intuitive Language Instruction

Figure 1: Counter-Intuitive Task Definition Examples

*states* that match the expected states. This metric is useful to check to what extent the model develops correct **partial understanding** in response to situational changes.

$$\text{State-EM} = \frac{\#(\text{Matched States})}{\#(\text{Queried States})} \quad (1)$$

**Step-EM** is a stricter metric than State-EM. It is the proportion of *steps* for which the full set of predicted states at that step have an exact match with the expected states, including all the truth values and the number of predicted states. This allows us to check whether the model can maintain **consistent** and **coherent** understanding over situational changes. This metric is also important given the sparsity of updates at each step, to ensure that models cannot achieve strong performance simply by copying previous states.

$$\text{Step-EM} = \begin{cases} 1 & \text{if Matched States=Ground Truth States} \\ & \qquad \text{=Predicted States} \\ 0 & \text{otherwise} \end{cases}$$

$$(2)$$

We illustrate the computation of these two metrics for the following simplified case (2 boxes and keys, synthetic language, no counterintuitive instructions):

```
9   [Instructions and some previous steps]
10  Question: NvSWxzvJb(jqC-0)=? B(bsS-0)=?
      NvSWxzvJb(jqC-1)=? B(bsS-1)=?
11  Correct answer: NvSWxzvJb(jqC-0)=True, B(
      bsS-0)=False, NvSWxzvJb(jqC-1)=False, B
      (bsS-1)=False
12  Model Answer: NvSWxzvJb(jqC-0)=True, B(bsS
      -0)=True, NvSWxzvJb(jqC-1)=False, B(bsS
      -1)=False
```

Example 4: Example for Metrics Computation

State-EM in this case is $3/4 = 75\%$ (only 3 out of 4 states are correct), while Step-EM is 0 because the step contains a incorrect state.

When computing these metrics, we find that at times the generated output does not strictly follow the given format from the in-context samples. Therefore, we use regular expressions to extract the truth values and corresponding states from model outputs. Details can be found in Appendix D.

**Existence of Shortcuts** It is clear that simple string automata plus a status tracking table should already be sufficient to solve every instance of our tasks. However, the simplicity of the task is part of its value: if models have a basic capacity to track and maintain environment states, this task should be straightforward. Nonetheless, we will see in the experiments below that ChatGPT still struggles to solve these tasks reliably, despite the existence of such simple solutions, indicating the presence of fundamental limitations in this class of capability.

### 3.4 Discussion: Synthetic Environment as Simulation for Real-World Application

At its core, our synthetic environment is a simplified simulation for real-world state-tracking tasks (usually in the form of slot-filling), a critical capability of dialogue systems / chatbots (Williams et al., 2014; Henderson et al., 2014; Wu et al., 2019). By prompting the model to update states, we are gradually giving the model more contextual information and testing whether ChatGPT, the state-of-the-art chatbot model, can closely follow users' prompts and keep track of the full interaction history.

Our work has important potential implications as the usage of LLMs continues to proliferate. Instructing LLMs to remember many initial states, operate over synthetic languages, and keep track of interaction history can be seen as an important step in eventually teaching a highly-capable agent to follow social norms and policies. Our initial set of environment states is similar to establishing basic rules about dos-and-don'ts at the beginning of human-AI conversations. The usage of synthetic languages is likewise similar to teaching AI agents about specific tones or styles of communication, terminologies, jargon, or perhaps even low-resource languages. Analyzing whether the model can keep

track of environment states can allow us to draw conclusions about the model's ability to follow instructions or rules. In this sense, our work also has implications with respect to recent trends of Constitutional AI or Rule-based/Principle-driven models (Bai et al., 2022; OpenAI, 2023; Sun et al., 2023), in which human social rules ("constitution") are explicitly written at the beginning of a prompt to align AI agents' behavior to human values. Our initial environment states are like constitution policies that AI agents should obey. The steps and queries in our environments are reminiscent of a situation in which certain policies can be allowed to be updated with user permission. For example, initially an AI agent may be programmed to try its best to answer every question and disallow overly conservative responses like refuse-to-answer—but under certain situations, the user could update with permission to the agent to refuse to answer for privacy, fairness or other social reasons.

As we will see in the experiments below, when more interactions occur, the model will gradually lose track of states, propagate errors, and even generate hallucinations, despite all updates falling within the input window. By design, a super-capable AI agent like ChatGPT should have the ability to read and use all information within the input window—but our results suggest that this is not the case. Our research thus calls for further study, and for caution when implementing a Constitutional AI approach.

## 4 Experiments: testing model sensitivity to instructions and situation change

We use our environment to test ChatGPT in its ability to track situation changes, and we report the results below. We also compare against the performance of additional chat models, which we find to underperform ChatGPT. Those results can be found in Appendix E.

We try 2-shot, 3-shot and 5-shot settings (in which 2, 3 and 5 example steps with enumerated states are provided). We use 50 samples with the number of additional steps randomly sampled from $\{1, \ldots, 10 - \#(\text{num-shots})\}$ for each setting. We follow the official OpenAI cookbook[2] to prepare the request and parse the response when interacting with ChatGPT API. Details are in Appendix C.

---

[2] https://github.com/openai/openai-cookbook/tree/main

### 4.1 Results

Results for these tests on ChatGPT are shown in Table 1. We can see a number of patterns.

**Failures on Step-EM** Under normal instructions, though the model achieves high State-EM (i.e., 90%), the Step-EM is generally much lower (up to 70% lower in 2-shot and $40 - 50\%$ lower in 5-shot). This indicates that although models are able to take advantage of the sparsity of the updates to get a majority of states correct, they are much less successful in accurately characterizing the entire environment state at a given step. As we will see in Section 5.2 State-EM may also be skewed by accidentally-correct states, and should in general be interpreted with some caution.

**ChatGPT has limited capability to understand and follow counterintuitive instruction.** From Table 1, we see that applying our counterintuitive instruction manipulation leads to significantly degraded performance. Especially on Step-EM, performance in most settings is $10 - 20\%$, which is much worse than with normal instructions ($> 20\%$, or even $> 30\%$ in most cases). This suggests that the model is indeed to some extent relying on standard language-to-logic mapping from pre-training, rather than fully understanding the instructions.

Despite this inconsistency, the model still shows some success with the negation or the new rule with flipped truth values, as it still manages to achieve $70 - 80\%$ State-EM. Though the State-EM values should be taken with caution, these accuracies are substantially stronger than would be expected from random guessing (50%) or completely ignoring the rule (0%), suggesting some level of capability in making use of the counterintuitive instructions.

**Effects of synthetic language** Use of synthetic language affects model performance, but not always in the predicted directions. Though performance is occasionally worse with synthetic language, it is more often *better* than with natural language. This suggests that the use of synthetic language may at times help the models to detach from unhelpful biases from pre-training, and rely more robustly on the in-context information. This follows the analysis presented in Liang et al. (2022).

**More in-context samples do not necessarily help.** We also see that even if we are already providing answers for approximately 50% of states (considering that we only have 10 boxes and 10 keys in the

| Normal Instruction | Step-EM / State-EM | | |
|---|---|---|---|
| | 2-shot | 3-shot | 5-shot |
| NL Functor + NL Argument | 22%/92% | 34%/93% | 36%/95% |
| SL Functor + SL Argument | 19%/92% | 22%/93% | 52%/96% |
| Counter-Intuitive Instruction (On NL) | | | |
| NL Functor + NL Argument | 10%/77% | 6%/75% | 0%/84% |
| SL Functor + SL Argument | 13%/89% | 20%/90% | 12%/90% |
| Counter-Intuitive Instruction (Truth Values Switching) | | | |
| NL Functor + NL Argument | 6%/72% | 2%/69% | 2%/79% |
| SL Functor + SL Argument | 19%/85% | 14%/87% | 12%/89% |

Table 1: Experiment results on ChatGPT Robustness check for state tracking in 10-box environment. Metrics here are presented in the format of "Step-EM / State-EM". We use 50 samples for each experiment setting.

environment, and each step will change exactly two states permanently within the dialog), the model does not make improvement in most cases.

# 5 Analysis of model performance

In the previous section we tested the capacity of ChatGPT in tracking and enumerating states within our environment, and we found that the model showed clear limitations. In this section, we analyze model performance further, to better understand the source of these limitations.

## 5.1 Tracing Errors: State Tracking over Steps

To understand why ChatGPT shows relatively poor performance, a straightforward way is reusing the instances created in Section 4, but querying for the environment states after each intermediate step to see where the errors emerge. Specifically, rather than only querying after the final step, we make queries at all steps (excluding "Step-0"), including those within the in-context example window (for querying each step $s$ in the in-context example window, in-context demonstrations of environment states are given only through step $s - 1$). We evaluate State-EM and Step-EM at every step. We refer to this test as **Intermediate State Probing**.

**Potential confounder: state complexity** When interpreting performance trajectory across increasing numbers of steps, a potential confounder is that performance may degrade simply because the set of environment states has become more complex, and not because there are too many steps of updates involved. To investigate this possibility, we also run a test in which we compress and skip $k$ of the early steps, and initialize in the state that would have resulted from those steps. We then test the trajectory of model performance on the subsequent

$n$ steps. If performance after $n$ steps in this setting is comparable to performance after $k + n$ steps in the previous setting, this will suggest that the degradation is indeed due to the complexity of the environment states. If, however, performance after $n$ steps in this setting is substantially better than performance at $k + n$ steps in the prior analysis, this suggests that the degradation in performance is due to failure to handle the growing number of steps.

| Normal Initialization | Step-EM / State-EM | |
|---|---|---|
| | 2-shot | 5-shot |
| NL Functor + NL Argument | 22% / 92% | 36% / 95% |
| SL Functor + SL Argument | 19% / 92% | 52% / 96% |
| Compressed Initialization | | |
| NL Functor + NL Argument | 46% / 95% | 60% / 97% |
| SL Functor + SL Argument | 48% / 95% | 54% / 97% |

Table 2: Compressed Initialization Testing experiment results for 10-box environment on ChatGPT. Metrics here are shown in the format of "Step-EM/State-EM". We use 50 samples with various number of steps for experiments.

The experiment results for the Intermediate State Probing and Compressed Initialization Test are shown in Figure 2 and Table 2, respectively. From these results we make the following observations:

**Degraded performance over steps.** We see in Figure 2 that with increasing number of situational changes, both State-EM and Step-EM degrade. This degradation is particularly true for Step-EM, which decreases dramatically as steps increase.

**State complexity does not explain the degradation.** Additionally, we see in Table 2 that skipping steps and starting with more complex initialization leads to improved performance, indicating that the

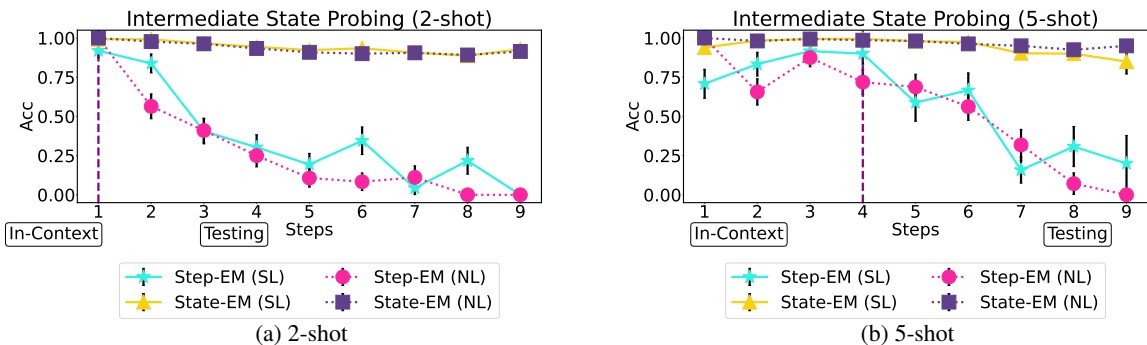

(a) 2-shot         (b) 5-shot

Figure 2: Results for Intermediate State Probing. Purple vertical lines indicate where in-context demonstrations (for steps prior to the test step) end.

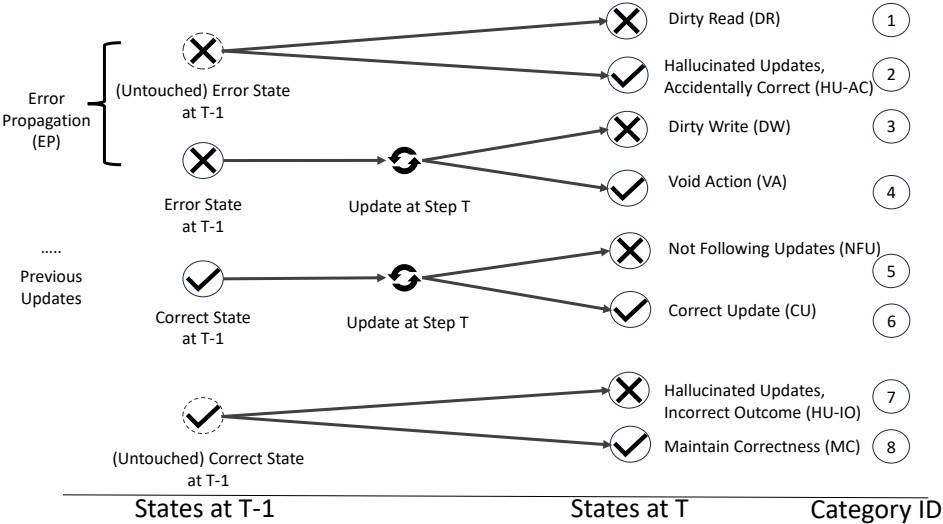

Figure 3: Diagram for fine-grained error analysis

degradation across steps is not attributable to state complexity alone.

**More in-context demonstration mitigates degradation.** In Section 4 we found that providing more in-context samples does not cause a significant improvement in the performance of the model at the final Test Query. With intermediate probing we gain a finer-grained picture of the impacts of number of in-context samples across steps. First, we see that when prompting with more in-context samples (5-shot vs 2-shot), there is less rapid degradation in model accuracy after the end of the in-context window. In the 2-shot (Figure 2a) case, at Step-3 (the second step preceded by a non-demonstration step), Step-EM quickly drops from 60% to 40%. By contrast, in 5-shot (Figure 2b), at Step-6, ChatGPT can still maintain a Step-EM value of 60%. This suggests that having more in-context samples does strengthen models' accuracy

in tracking state changes—but only temporarily or over a limited number of steps.

Though we see that having more demonstrations can mitigate degradation after the demonstrations end, when we look within the in-context sample window itself, we see that on steps that directly follow in-context demonstrations (Step-1 for 2-shot, Step-1,2,3,4 for 5-shot), the model's performance does not monotonically increase in response to the accumulating demonstrations. A similar phenomenon is also discovered in other few-shot reasoning benchmark works (Suzgun et al., 2022; Liang et al., 2022), despite the fact that in traditional fine-tuning, usually, more training instances yield better generalization ability. This suggests that although adding more demonstrations can briefly mitigate loss of accuracy, it does not straightforwardly translate to gains in accuracy.

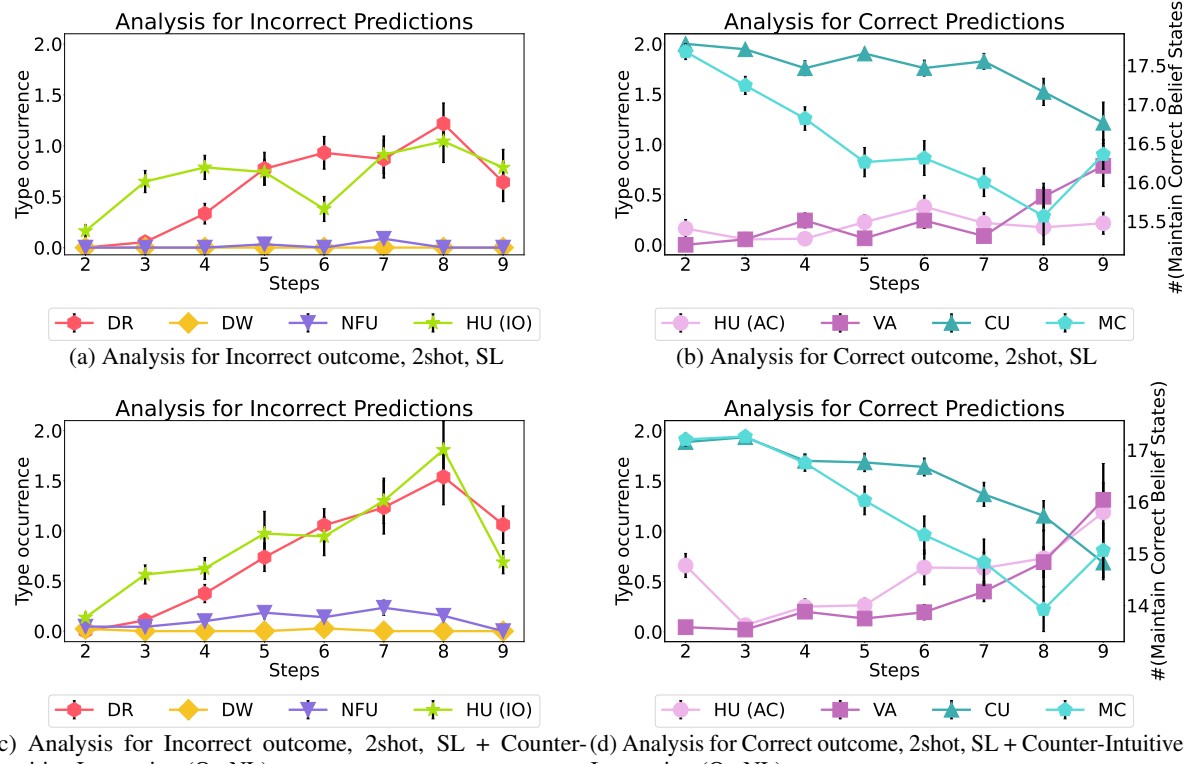

(a) Analysis for Incorrect outcome, 2shot, SL

(b) Analysis for Correct outcome, 2shot, SL

(c) Analysis for Incorrect outcome, 2shot, SL + Counter-Intuitive Instruction (On NL)

(d) Analysis for Correct outcome, 2shot, SL + Counter-Intuitive Instruction (On NL)

Figure 4: Fine-Grained Error Analysis for Logical Inconsistency

**Interim discussion** These patterns of degradation over time occur, in both NL and SL settings, despite the fact that ChatGPT can read the full dialogue in its input window. This suggests that ChatGPT cannot effectively utilize the full information in its input window, and that claims about maximum input length capabilities (e.g., ChatGPT can model 4k tokens as introduced in the official announcement (OpenAI, 2022)) should be taken with a grain of salt.

### 5.2 Fine-grained analysis of update patterns

In the above section, we studied the trajectory of model performance as the number of steps increases, finding evidence that ChatGPT degrades in state tracking with increased number of steps. In this section, we do a finer-grained analysis of the update dynamics in these experiments, in order to examine more closely the causal factors leading to both erroneous and accurate predictions. For the purpose of this analysis, we define categories of state transitions, summarized in Figure 3. These categories allow us to analyze the relationships between states at the analyzed step and the corresponding prior state, both when updates should be made to those states and when they should not.

The experiment results are shown in Figure 4 (we only show 2-shot SL, for reasons of space).

Examining first the patterns for states in which models make incorrect predictions, we see that the rise in errors is driven by states that should be untouched at that step. We see that as steps increase there are rapid increases in both Dirty Read (DR) transitions, where models retain a previous error, and Hallucinated Update (HU-IO) transitions, where models change a state from correct to incorrect despite there being no change to that state in the step description. These patterns indicate that the rise in errors over time can be attributed both to retention and propagation of errors from previous steps, but also to *failures* in retaining the memory of a previous step that should not change.

Examining now the transitions associated with correct model predictions, we see that over time there is noteworthy decrease in Correct Update (CU) cases—however, there is a much more dramatic decrease in Maintain Correctness (MC) cases, indicating that the model increasingly fails to retain the memory of previously correct states. Over time we also see, particularly in the case of counterintuitive instructions, a rise in Accidentally Correct (HU-AC) cases, in which the model switches from

an incorrect state back to the correct state, despite the fact that no update to that state was described in the step. Both of these patterns are indicative of memory limitations and susceptibility to random noise in changes to enumerated states.

These results yield several conclusions:

**ChatGPT has non-persistent in-context memory.** A recurring observation above is that many of the model errors that increase over time can be attributed to limitations in retaining states in memory—and in fact, some states marked as correct also reflect accidental correctness arising due to similar failures to retain prior states.

**States can also be retained, but potentially by chance.** In addition to memory retention failures, we also see propagation of errors between steps—which in theory is indicative of successful memory retention, by contrast to the retention failures cited above. However, considering the prevalence of hallucinated updates, and the limited options for state values, we can expect that at least some of these retained updates in fact occur by chance.

**Couterintuitive instruction exacerbates non-robust behavior.** As we mentioned above, the drop in correct CU updates is more dramatic—and the rise in spurious correct updates HU more substantial—in the case of counterintuitive instructions. This suggests that the inability of the model to rely on memorized language-to-logic mappings generally reduces the model's ability to execute and maintain correct state updates.

## 6   Conclusion

In this paper, we propose a novel synthetic testing environment for testing situational understanding capabilities, which we apply to test ChatGPT, the state-of-the-art chatbot. We instruct ChatGPT to process a series of sparse environment updates across time in a dialogue history. With careful environmental designs, we reduce the possibility of data contamination and other artifacts typically introduced by traditional probing methods. We find that despite the simplicity of our task, and even with ChatGPT having full access to the complete dialogue history within the input window, the model fails to retain coherent and correct environment states over time. Further analysis suggests that this failure is largely because ChatGPT does not have persistent in-context memory, and is susceptible to

hallucinated updates. These findings indicate overall that ChatGPT does not have robust situational state tracking ability.

Our proposed synthetic environment and the findings that it generates can have noteworthy real-world implications. First, it can diagnose the potential limitations of current chatbot systems in multi-round interactions. Second, our findings also reflect a potential problem for the model's ability to follow instructions and remain consistent with rules/norms established in its context, which is especially important for responsible AI safety and human-AI alignment research.

## Acknowledgements

We are grateful for the insightful discussion with Chih-chan Tien (UChicago), Kanishika Misra (UT Austin), Hao Zhu (CMU) and Zhaofeng Wu (MIT) at the early stage of this work (names are not listed in particular order). We also thank the anonymous EMNLP reviewers and chairs for providing insightful and constructive feedback to make this work more solid.

## Limitations

In this work, we propose a controlled synthetic environment to investigate ChatGPT's situational understanding ability. While we believe our synthetic environment has important real-world implications, as we discussed in Section 3.4, for certain real-world applications our findings may not apply. Another limitation is that we only focus on the evaluation of ChatGPT as the state-of-the-art chatbot model (at least in terms of mainstream media coverage). There are other commercial chatbot models that could show stronger performance on our tasks, as they may have more complicated system designs (e.g., multi-module systems as in BlenderBot 3 (Shuster et al., 2022)) that could be better at dealing with multi-round dialogue history and extremely long inputs. As we do not have sufficient budget or open access to test many such systems, we leave a comprehensive benchmark evaluation of situational understanding ability for Chat-Tuned LLMs for future work. Our experiments closely follow the OpenAI official cookbook for interacting with ChatGPT, but it is possible that there could be more optimal prompts to fully unlock the capability of ChatGPT.

There are many other synthetic environments like TextWorld (Côté et al., 2019) that may be pro-

grammed to do situational testing as in our work (though it may not be easy to assert full controls), and it is would be interesting to establish whether in different environments we can still draw the same conclusions. Our work mainly focuses on our proposed environment as a case study, but we plan to extend our testing framework to other environments in the future.

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

## A  Distractor on Action

In addition to instruction understanding, the model should also learn to correctly understand and select key information every time a new step is taken. To examine the robustness of this capability, in addition to the inclusion of normal step descriptions, we also include distractor conditions (Shi et al., 2023), in which we add one randomly selected sentence from the Brown Corpus (Kučera and Francis, 1967) as a distractor. An example of applying such distractor is shown in Example 5. The experiment results are shown in Table 3.

```
13 │ Step-1: Open jqC-3 and retrieve bsS-2. It
   │     is a nice day!
14 │ Question: NvSWxzvJb(jqC-0)=False...
```

Example 5: Action Distractor Example

| Normal Instruction | Step-EM / State-EM | |
| --- | --- | --- |
| | 2-shot | 5-shot |
| NL Functor + NL Argument | 22% / 92% | 36% / 95% |
| Synthetic Functor + Synthetic Argument | 19% / 92% | 52% / 96% |
| Actions w/ Distractor | | |
| NL Functor + NL Argument | 18% / 91% | 44% / 96% |
| Synthetic Functor + Synthetic Argument | 34% / 93% | 50% / 96% |

Table 3: Action understanding experiment results for 10-box environment on ChatGPT. Metrics here are shown in the format of "Step-EM/State-EM". We use 50 samples with various number of steps for experiments.

From Table 3, we can see after adding distractors, the State-EM performance does not degrade as much as in counter-intuitive instructions, though Step-EM performance degrades a bit (but not consistently). These findings hold both in NL and synthetic languages. This suggests that ChatGPT does have the ability to understand the interaction happened at each step and can pick out useful information.

We also see that when adding distractors on actions, within the 2-shot condition, Step-EM in synthetic language environment is better than the one in NL environment. This is probably because the usage of synthetic language helps the model better distinguish useful information as they look very different from synthetic languages. However, when there are more in-context samples, the model will gradually learn to extract useful information at each step and the usage of synthetic language does not help that much.

## B  Partial Usage of Synthetic Languages

We can do a more fine-grained usage of synthetic langauges – we can choose to apply only on functors (e.g., "Opened") or arguments (e.g., "BOX-1"). The full set of experiment results is shown in Table 4. We notice that the results are very similar to the full usage of synthetic languages so to save space, we move the results in Appendix.

## C  Interaction with OpenAI API and Prompt Format

According to the official OpenAI cookbook,[3] to send requests to ChatGPT, it is advised to first add a "system message" to ChatGPT (We use "You are a helpful assistant" as this is one of the most used system messages). Then there are two typical ways[4] to send the prompt, if it contains in-context samples:

**Traditional Format**  Just put all your prompt contents in one-round interaction (as in prompting GPT-3-Davinci series models), as shown in Example 6:

```
15 │ response = openai.ChatCompletion.create(
16 │     model=...,
17 │     messages=[
18 │         {"role": "system", "content": "You
   │     are a helpful assistant."},
19 │         {"role": "user", "content": "[
   │     Instruction + In-Context Samples]"},
20 │     ],
21 │     ...,
22 │ )
```

Example 6: Traditional Input Format

**Faked Multi-Round Format**  Another way is to synthesize a fake multi-round conversation and pretend the answers for in-context samples are generated by ChatGPT, as shown in Example 7:

```
23 │ response = openai.ChatCompletion.create(
24 │     model=...,
25 │     messages=[
26 │         {"role": "system", "content": "You
   │     are a helpful assistant."},
27 │         {"role": "user", "content": "[
   │     Instruction]"},
28 │         {"role": "user", "content": "[
   │     Sample-1-Input]"},
29 │         {"role": "assistant", "content": "[
   │     Sample-1-Ground-Truth-Answer]"},
```

---

[3] https://github.com/openai/openai-cookbook/blob/main/examples/How_to_format_inputs_to_ChatGPT_models.ipynb

[4] As of the time the project is initialized. OpenAI may change a bit in the documentation and examples in the GitHub repository.

| Normal Instruction | Step-EM / State-EM | | |
|---|---|---|---|
| | 2-shot | 3-shot | 5-shot |
| NL Functor + NL Argument | 22%/92% | 34%/93% | 36%/95% |
| Synthetic Functor + NL Argument | 14%/90% | 20%/93% | 30%/94% |
| NL Functor + Synthetic Argument | 44%/95% | 30%/94% | 74%/98% |
| Synthetic Functor + Synthetic Argument | 19%/92% | 22%/93% | 52%/96% |
| Counter-Intuitive Instruction (On NL) | | | |
| NL Functor + NL Argument | 10%/77% | 6%/75% | 0%/84% |
| Synthetic Functor + NL Argument | 18%/88% | 10%/88% | 14%/90% |
| NL Functor + Synthetic Argument | 20%/81% | 10%/78% | 8%/88% |
| Synthetic Functor + Synthetic Argument | 13%/89% | 20%/90% | 12%/90% |
| Counter-Intuitive Instruction (Truth Values Switching) | | | |
| NL Functor + NL Argument | 6%/72% | 2%/69% | 2%/79% |
| Synthetic Functor + NL Argument | 10%/85% | 10%/84% | 6%/87% |
| NL Functor + Synthetic Argument | 8%/71% | 8%/73% | 0%/83% |
| Synthetic Functor + Synthetic Argument | 19%/85% | 14%/87% | 12%/89% |

Table 4: Experiment results on ChatGPT Robustness check for state tracking in 10-box environment. Metrics here are presented in the format of "Step-EM / State-EM". We use 50 samples with various number of steps for experiments.

```
30        {"role": "user", "content": "[
   Sample-2-Input]"},
31        {"role": "assistant", "content": "[
   Sample-2-Ground-Truth-Answer]"},
32        ...
33      ],
34      ...,
35  )
```

Example 7: Faked Multi-Round Input Format

| Normal Instruction | State-EM, 2-shot | |
|---|---|---|
| | ChatGPT | Text-Davinci-003 |
| NL Functor + NL Argument | 92% | 96% |
| Synthetic Functor + NL Argument | 90% | 94% |
| NL Functor + Synthetic Argument | 95% | 94% |
| Synthetic Functor + Synthetic Argument | 92% | 88% |

Table 5: Experiment Results Comparing ChatGPT and Davinci-003. Metrics here are State-EM. We use 50 samples with 2-shot setting and various number of steps for experiments.

We do a prior study on experimenting with these two formats and find they give pretty similar Step-EM and State-EM performance in fully NL and fully Synthetic Language settings (usually the performance difference is less than 5%). But faked multi-round format would insert many more tokens (leading to a higher cost) in the requests and can make the OpenAI server reject to respond, leading to low response rates. Therefore, to save budgets, in the main text of the paper, we only report the results using the traditional format. Another reason is that to see how much our findings on ChatGPT can generalize to other models, we also replicate some of our experiments when comparing with other models in Appendix E. Some models there may not support faked multi-round input format.

For response parsing, we just follow the instruction to parse the JSON-style response described in the same notebook and obtain the model output. For decoding parameters, we just follow default OpenAI API setting.

# D  Post-Processing to Extract Answers from OpenAI API Responses

We note that ChatGPT can return answers not strictly follow the given format. So we do a lightweight postprocessing mainly using regular expressions to make sure we parse ChatGPT results appropriately. Specifically, we first clean out the space and newline characters at the beginning and end of the answer. We then use the following regular expression to match all logical statements and extract groups of functors, arguments and truth values:

```
36   ([a-zA-Z0-9]+)\(([a-zA-Z0-9]+-\d)\)=(
   True|true|False|false)
```

# E  Comparison with other models

To clarify whether our observations can be generalized to other popular models, we choose one strong open-sourced competitor chatbot (according to ChatArena (Zheng et al., 2023)): Vicuna-13b—and we also compare against the performance of GPT-3-davinci-003 (it is believed that ChatGPT is a variant fine-tuned over GPT-3).

**Comparison between Vicuna-13B** On ChatArena (Zheng et al., 2023), Vicuna-13B (Chiang et al., 2023) is voted to be the most close-to-ChatGPT chat models based on LLAMA (Touvron et al., 2023). We feed the same input to Vicuna and find the outputs are hard to parse and often incomplete. Even when there are no counter-intuitive instructions and we give 5 in-context samples, the best observed State-EM (in SL environment) is only 32%. Compared with ChatGPT (94%), there seems a big gap on the capability of situational understanding for open-source models.

**Comparison between GPT-3.5 and ChatGPT** Due to the budget limit and context length limit of GPT-3.5, we only compare ChatGPT performance with GPT-3.5 (Davinci-003) on 2-shot. The experiment results are in Table 5. We can see Davinci-003 achieves a similar performance as ChatGPT.