# OpenReview forum: "Can You Follow Me? Testing Situational Understanding for ChatGPT"
_EMNLP/2023/Conference — EMNLP 2023 Main_

### Official Review · Reviewer_q8HG · 2023-07-31

**Soundness:** 5

**Excitement:**

4: Strong: This paper deepens the understanding of some phenomenon or lowers the barriers to an existing research direction.

**Paper Topic And Main Contributions:**

The paper describes an evaluation setup for LLMs that tests their ability to keep track of situational changes in a dialog. The LLMs are tasked to keep track of objects whose state can change with certain actions at each step and is then asked to present the new set of states. The paper describes the setup in detail and presents detailed discussions of the evaluation run with ChatGPT, showing its shortcomings in tracking dialog states.

The main contributions are
- the design and description of the evaluation setup that can be applied to any language or dialog model
- the application of the setup to ChatGPT
- the analysis and discussion of the results that show that the model has trouble to remain consistent over time

**Questions For The Authors:**

A: I do not fully understand the difference between the two evaluation metrics, Step-EM and State-EM, could you please give an example?

**Reasons To Accept:**

- The presented evaluation setup is carefully designed. The task is well chosen and argued for and measures are taken to minimize the tested models' reliance on common patterns. For examples, random string are used for state predicates. The task is both simple enough to test basic capabilities and complex enough to provide insight into a fundamental dialog capability.

- The setup and results are discussed in detail, providing value to others who want to build on this work and understand the phenomenon.

- The results are interesting and detailed, showing how correct and incorrect states happen at different stages of the tested dialogs.

- The paper is well-written and structured. I have some comments in the presentation section below.

**Reasons To Reject:**

- The paper does not mention whether the code would be made public in order for others to replicate and evaluate other LMs and extend the settings.

**Reproducibility:**

4: Could mostly reproduce the results, but there may be some variation because of sample variance or minor variations in their interpretation of the protocol or method.

**Reviewer Confidence:**

4: Quite sure. I tried to check the important points carefully. It's unlikely, though conceivable, that I missed something that should affect my ratings.

**Typos Grammar Style And Presentation Improvements:**

- Table 1: I find the tight format of the numbers hard to read. You seem to have enough space to add spaces around the slash. Right-aligning the numbers makes the single digits stand out more.

- Section 4.1, Failures on Step-EM: I think you mean that the results are 70 percentage points lower, not 70% lower

- Section 4.1, Effects of synth langugae: "better then" -> "better than"

---

> ### Author Rebuttal · Authors · 2023-08-28
>
> We thank the reviewer for the thoughtful feedback. Below we address the concern and question raised.
>
> **Re: Code Release**
>
> We will release all experimental code in the formal version to help others build the test environment and generate test examples.
>
> **Re: Two Metrics Design**
>
> Thank you for pointing out this confusion. We will clarify here, and we will add clarification in the revised version as well. Basically, State-EM assesses the percentage of states within a step that the model gets correct. So for this measure, at a given step, the model gets credit for all states that it got correct, and it is penalized for all states that it missed. By contrast, Step-EM evaluates the model’s ability to maintain a fully consistent and correct situational state – so for this measure, at a given step, the model only gets credit if it gets all states in that step correct. If any states are incorrect, the model does not get credit for that step.
>
> To illustrate now with an example, consider the following made-up case:
>
> [Instructions and some previous steps]
>
> Question: NvSWxzvJb(jqC-0)=? B(bsS-0)=?
>
> Correct answer: NvSWxzvJb(jqC-0)=True, B(bsS-0)=False
>
> Model Answer: NvSWxzvJb(jqC-0)=True, B(bsS-0)=True
>
> State-EM for this case: ½ = 50%
>
> Step-EM for this case: 0 (as there contains one error state)
>
> **Regarding Presentations, Typos and Styles**
>
> Thank you for catching these! We will fix these in the revised version.

---

### Official Review · Reviewer_rJkr · 2023-08-04

**Soundness:** 4

**Excitement:**

3: Ambivalent: It has merits (e.g., it reports state-of-the-art results, the idea is nice), but there are key weaknesses (e.g., it describes incremental work), and it can significantly benefit from another round of revision. However, I won't object to accepting it if my co-reviewers champion it.

**Paper Topic And Main Contributions:**

The paper presents a simple synthetic testing environment for testing the ability of situational understanding (SU) of large language models. Specifically, they use ChatGPT as a case study. The authors propose a synthetic box-moving environment like TextGame (Li et al., 2021) to assess the models’ ability to track and enumerate environment states and analyze the causes for performance patterns. They find that ChatGPT struggles to retain coherent environment states over time, mainly due to non-persistent in-context memory and susceptibility to hallucinated updates. The authors also conduct a fine-grained analysis of the updated dynamics to illustrate the causal factors of error predictions. The paper contributes to the understanding of the limitations of the current ChatGPT model and highlights the need for improved situation tracking abilities.


**Reasons To Accept:**

1) The paper is well-written and properly organized.

2) The paper introduces a novel synthetic environment for testing situational understanding, which provides a good perspective for evaluating large language models.

3) The fine-grained error analysis provides a great understanding of the dynamics of the model performance, illustrating how correct or wrong transitions were made. It is particularly interesting to see the accidental correctness goes up as steps increase.

**Reasons To Reject:**

1) Some implementation details of the synthetic environment are missing. For example, how do you design the step instructions for each of the 50 samples? Do they come from random permutations?

2) Reproducibility might be an issue. Are you going to release all your prompts so others can replicate the whole environment and results?

3) The difficulty of the task, especially given two different evaluation metrics, is not clear to me. The authors compare ChatGPT with other chatbots. But how does it compare to the fine-tuned models? If large language models such as T5 were trained on this task, will they perform better or worse than ChatGPT?

**Reproducibility:**

2: Would be hard pressed to reproduce the results. The contribution depends on data that are simply not available outside the author's institution or consortium; not enough details are provided.

**Reviewer Confidence:**

4: Quite sure. I tried to check the important points carefully. It's unlikely, though conceivable, that I missed something that should affect my ratings.

**Typos Grammar Style And Presentation Improvements:**

I suggest using 10 instead of 5 in Example 1 to avoid confusion.

---

> ### Author Rebuttal · Authors · 2023-08-28
>
> We're grateful for the constructive feedback provided by the reviewer. In the following section, we address the mentioned concerns and questions in turn.
>
> **Re: Missing Implementation Details**
>
> Thanks for raising this concern. For designing step instructions for each of the 50 samples, we sample the number of steps randomly from a uniform distribution U(1, 10). We then keep instructions almost entirely the same across samples, except that in the Synthetic Language settings the state predicates are defined randomly, and therefore vary between samples (e.g., in one instance we use “NvSWxzvJb(jqC-0)=True“ to refer to the box jqC-0 has been opened (“NvSWxzvJb”-ed), while in the other instance, we use “Abc(bb-0)=True” to represent the box bb-0 has been opened (“Abc”-ed) ). We will add these clarifications in the revised version.
>
> **Re: Reproduction Issues on the release of all prompts**
>
> We will release all prompts and API responses from ChatGPT for reproduction in the formal version. We will also release the environment building codes for users to build different testing environments and generate test examples.
>
> **Re: The difficulty of the task, especially given two different evaluation metrics, is not clear to me. The authors compare ChatGPT with other chatbots. But how does it compare to the fine-tuned models? If large language models such as T5 were trained on this task, will they perform better or worse than ChatGPT?**
>
> **Regarding Task difficulty**: In Section 3.3, we briefly discuss the task difficulty and the design consideration in the “Existence of Shortcuts” paragraph: we believe this task is in principle very simple, as long as the tested agent has the fundamental situational understanding ability – note that we only need a string automaton and a state-tracking table to achieve 100% EM on both metrics. However, we carefully design our testing environment and control for confounders, so that the task is *not* simple for models that lack this situational understanding ability – in particular, we work to avoid models being able to solve the task via easier, shallower solutions that don’t require the target situational understanding ability. In this way, our tests allow us to draw rigorous conclusions about situational understanding capabilities.
>
> **Regarding Comparison against fine-tuned T5**: We do expect that we could see improved performance by actively fine-tuning a model (like T5) to do our task, but this would be somewhat orthogonal to our purpose in this submission, which is to test the existing situational understanding capabilities of these state-of-the-art chat models out of the box. An important component of our test is that it taps into models’ fundamental situational understanding capabilities as they are, by providing a small number of demonstrations of the task format, but otherwise not doing any fine-tuning of the model to map between these specific inputs and outputs.
>
> As for a scenario in which we compare against T5 in the few-shot setting that our test is designed for: we do not have results for this comparison on hand, but we speculate that because T5 does not go through the RLHF process, it may be difficult for T5 models to follow instructions and handle multi-round interactions in our environment – so we hypothesize that T5 will perform much worse than ChatGPT on our task as designed.
>
> As a final note, in Appendix E we also compare ChatGPT with Davinci-003, which is reported to have the same base model as ChatGPT but is optimized more using multi-task instruction tuning, similar to T5 (See [OpenAI official Model Index](https://platform.openai.com/docs/model-index-for-researchers) for reference). We find that this model behaves similarly to ChatGPT, though at a much higher cost. So we focus on the evaluation of ChatGPT in this paper, but we ensure that it is straightforward to extend our evaluation to T5 or other fine-tuned models, as desired, since our evaluation is agnostic as to the model being prompted.

---

### Official Review · Reviewer_xaGB · 2023-08-05

**Soundness:** 4

**Excitement:**

3: Ambivalent: It has merits (e.g., it reports state-of-the-art results, the idea is nice), but there are key weaknesses (e.g., it describes incremental work), and it can significantly benefit from another round of revision. However, I won't object to accepting it if my co-reviewers champion it.

**Paper Topic And Main Contributions:**

### Topic
This paper mainly focuses on the ability to situational understanding of chatbots, mainly ChatGPT. The paper proposes a novel synthetic environment for controlled testing of models’ ability to track and enumerate environment states, which reflects an agent's capacity to ground language in situations and comprehend situational changes. The research sheds light on the limitations that natural language models face, particularly in understanding situational changes and serving as human-like agents to accomplish real-world tasks.

### Contributions
The authors state that ChatGPT fails to retain prior states in memory and has susceptibility to spurious hallucinated updates, and concluded that chatGPT cannot currently track environmental changes.

**Reasons To Accept:**

1. The paper reveals the limitations of natural language models in understanding situational changes and becoming human-like agents to accomplish real-world tasks; There is a discussion about the risk of trust in ChatGPT's impressive conversation performance; The findings have potential implications for teaching highly effective agents to follow social norms and policies.

**Reasons To Reject:**

In my opinion, the contribution of this paper is quite incremental:

0. LLMs gradually forget the conversation history, limited on the one hand by the length of its hard prompt and on the other by the length of the window used in its pre-training. This is what we already know, so the idea that comes out of the paper doesn't seem interesting.

1. The paper does not explore why the model has these limitations, and lacks in-depth explanation and exploration of these situations.

2. The synthetic environmental tests mentioned in the paper can be interfered with in some cases, and there are some potential confounding factors that could lead to conclusions.

3. The research focus of this paper is mainly on the ChatGPT model, and there is a lack of evaluation of other chat models, so the discussion on the generalization ability of the overall situational understanding ability of the chat model is limited.

4. The paper does not provide specific suggestions and methods on how to solve or improve the situational understanding of the chat model.

**Reproducibility:**

5: Could easily reproduce the results.

**Reviewer Confidence:**

4: Quite sure. I tried to check the important points carefully. It's unlikely, though conceivable, that I missed something that should affect my ratings.

---

> ### Author Rebuttal · Authors · 2023-08-28
>
> We appreciate the insightful comments from the reviewer. We have responded to the issues presented below.
>
> **Re: Conversation History Forgetting, Input Length Mismatch from Training Time**
>
> Thanks for pointing this out. We agree that the length extrapolation issue (i.e., that performance degrades when testing on inputs of longer length compared with the training phase) is a well-known problem for LLMs. However, in this paper, we focus on situations in which test inputs are of the same length as those the model is trained to receive. We make special efforts to control input length *not* to be a confounder in our experiments.
>
> First, with regard to the input length that the model can handle, according to the OpenAI ChatGPT [official API](https://platform.openai.com/docs/guides/gpt/managing-tokens) and [most recent release page](https://openai.com/blog/function-calling-and-other-api-updates), for the current ChatGPT version (“gpt-3.5-turbo”), 4k context length is within the model capability and the model should be able to handle it properly. To avoid the confounder effect of length extrapolation, we carefully followed [OpenAI CookBook](https://github.com/openai/openai-cookbook/blob/main/examples/How_to_count_tokens_with_tiktoken.ipynb) to count tokens using the `tiktoken` package, and we also controlled the length of our inputs (including our instructions – note that our instructions are much simpler compared with prompts in InstructGPT [1], which can easily go beyond 1k tokens) plus our expected outputs to ensure that we avoid going beyond the recommended context length 4k (e.g., via controlling the number of rounds we prompt ChatGPT).
>
> However, we find that even within the designated 4k input length, where the length extrapolation problem should not apply, we clearly find that ChatGPT will gradually lose track of situational changes and maintain broken internal states. We believe this is a substantive new finding that is different from previous works, and users should be made aware of this risk. (A contemporary work is “Lost in the Middle” [2], which was released after our EMNLP submission and takes a different approach to study a similar problem that we do, while focusing more on positional bias).
>
> **Re: The paper does not explore why the model has these limitations and lacks an in-depth explanation and exploration of these situations.**
>
> We respectfully disagree with the criticism that the paper lacks explanation and exploration of the situational understanding problems. First, in Section 5.1, we do interactive debugging to prompt each intermediate step for failure situations, and we find out 1) the model performance degrades over steps; 2) state complexity does not explain performance degradation; 3) more in-context samples help mitigate the performance drop problem in muti-round interactions. Then, in Section 5.2, we do a fine-grained error analysis to further illustrate the problem of error propagation, failure to maintain consistency, and the increasing risk of hallucination to gain better performance numbers. We believe that the insights from these explorations can help both in informing users about possible reasons for an undesired chat experience, and also ultimately in helping to build a more controllable, interactive, and trustworthy chat model in the future, by helping to zero in on the circumstances and characteristics of model failures.
>
> **Re: Confounders in our synthetic environmental tests**
>
> We appreciate the reviewer’s attention to this concern. We fully agree with the importance of controlling for confounders, and we have carefully designed our experiments to exert this control to the best of our ability. A non-exhaustive list of confounders we have addressed includes 1) as stated in the first item in this response, we carefully control the input length to avoid the length extrapolation problem; 2) we do a natural language - synthetic language comparison to reduce data contamination problems; 3) we do a counterfactual verification (Section 5.1) to examine whether state complexity would interfere with the model performance drop instead of solely multi-round interactions. These careful experimental designs are part of the main value of our work.
>
> If the reviewer could specify any particular confounding variables they believe were not controlled in our experiments, we would greatly appreciate the additional insight.
>
>
> **Re: The research focus of this paper is mainly on the ChatGPT model, and there is a lack of evaluation of other chat models, so the discussion on the generalization ability of the overall situational understanding ability of the chat model is limited.**
>
> As stated in Appendix E, we also compare with Vicuna-13B [3], which was, at our submission time, an open-sourced state-of-the-art (in terms of ChatArena Benchmark [4]) chat model via fine-tuning on LLAMA [5]. Our experiment results reveal that this model has more significant problems in understanding situational changes compared with ChatGPT, even in simplified environmental settings (e.g., in a 5-state environment, where ChatGPT can easily achieve 94% State-EM but Vicuna only achieves 32% and can produce very hard-to-understand outputs). We also compare with the non-chat model OpenAI Davinci-003 by giving the same prompts and we find the performance number is similar to ChatGPT. Therefore, we focus our evaluation mainly on ChatGPT to expose the situational understanding limitation for the strongest state-of-the-art Chat model. We recognize that there also exist commercial model-based Chat systems that may be able to use memory modules to handle our tasks, but as we do not have access to APIs for these models, we leave the benchmarking on these models for future work, and make a note about this in the Limitations section. We believe that our environmental tests are general enough for testing any chat models, laying a strong foundation for future comprehensive benchmarking works.
>
> **Re: The paper does not provide specific suggestions and methods on how to solve or improve the situational understanding of the chat model.**
>
> In this paper, our main focus is to expose these situational understanding issues in state-of-the-art chat models (i.e., ChatGPT), and to provide thorough in-depth analysis, providing insight and explanations with respect to model failures. We do provide some insights into how one might solve or improve the situational understanding of the chat model (e.g., via giving more in-context samples, or perhaps adding a memory module could help as our task can be solved by string automata + lookup tables), but proposing major methodological innovation goes beyond the scope of the paper. We believe, however, that our findings and evaluation can help model developers better design a more capable chat model with improved situational understanding ability, by assisting in pinpointing where and how these types of model failures are arising.
>
> References:
>
> [1] Ouyang, Long, et al. "Training language models to follow instructions with human feedback." Advances in Neural Information Processing Systems 35 (2022): 27730-27744.
>
> [2] Liu, Nelson F., et al. "Lost in the middle: How language models use long contexts." arXiv preprint arXiv:2307.03172 (2023).
>
> [3] Chiang, Wei-Lin, et al. "Vicuna: An open-source chatbot impressing gpt-4 with 90%* chatgpt quality." See https://vicuna. lmsys. org (accessed 14 April 2023) (2023).
>
> [4] Zheng, Lianmin, et al. "Judging LLM-as-a-judge with MT-Bench and Chatbot Arena." arXiv preprint arXiv:2306.05685 (2023).
>
> [5] Touvron, Hugo, et al. "Llama: Open and efficient foundation language models." arXiv preprint arXiv:2302.13971 (2023).

---

### Meta-Review · Area_Chair_U2P7 · 2023-09-20

**Recommendation:** 5

**Metareview:**

The paper at pointer delves into situational grasp within LLMs, emphasizing the ChatGPT model. Critical facets of the study include:

- The introduction of a synthetic environment akin to TextGame (as cited in one of the reviews), aimed at examining models' capability to chronicle and enumerate environment states.

- An extensive evaluation of ChatGPT's limitations in maintaining consistent environment states over extended periods attributed to transient in-context memory and proneness to hallucinated updates.

- Detailed error analyses reveal how LLMs transition from one state to another, encompassing correct and incorrect state transitions.

**Reasons to Accept**

- The paper uncovers the inherent limitations of LLMs, particularly ChatGPT, in tracking situational changes, emphasizing the challenges of LLMs in emulating human-like contextual comprehension and interactions.
- The synthetic environment introduced offers a novel paradigm to evaluate the situational understanding of LLMs. This environment sheds light on the intricate dynamics governing model performance, presenting invaluable insights into the nuances of state transitions.
- An in-depth discussion, spanning the risks associated with an unnecessary dependency on ChatGPT's conversational prowess to the broader importance of designing agents that align with societal norms and policies, lends a holistic perspective to the paper.

**Reasons to Reject**

- Reproducibility Concerns: The paper needs to include essential implementation details of the synthetic environment. Furthermore, there needs to be more clarity about the release of necessary prompts and the step instructions' design, making it challenging for others to replicate the results and potentially question the study's validity.
- Limited Scope: The research predominantly concentrates on ChatGPT, neglecting a broader examination of other chat models. This limited focus restricts the paper's applicability and generalizability in the broader context of chatbot situational understanding.
- Evaluation Metrics Ambiguity: The difference between specific evaluation metrics, specifically Step-EM and State-EM, must be articulated, leading to potential misunderstandings about the study's conclusions.

---

### Decision · Program_Chairs · 2023-10-07

**Decision:**

Accept-Main

**Comment:**

The paper at pointer delves into situational grasp within LLMs, emphasizing the ChatGPT model. Critical facets of the study include:

- The introduction of a synthetic environment akin to TextGame (as cited in one of the reviews), aimed at examining models' capability to chronicle and enumerate environment states.

- An extensive evaluation of ChatGPT's limitations in maintaining consistent environment states over extended periods attributed to transient in-context memory and proneness to hallucinated updates.

- Detailed error analyses reveal how LLMs transition from one state to another, encompassing correct and incorrect state transitions.

**Reasons to Accept**

- The paper uncovers the inherent limitations of LLMs, particularly ChatGPT, in tracking situational changes, emphasizing the challenges of LLMs in emulating human-like contextual comprehension and interactions.
- The synthetic environment introduced offers a novel paradigm to evaluate the situational understanding of LLMs. This environment sheds light on the intricate dynamics governing model performance, presenting invaluable insights into the nuances of state transitions.
- An in-depth discussion, spanning the risks associated with an unnecessary dependency on ChatGPT's conversational prowess to the broader importance of designing agents that align with societal norms and policies, lends a holistic perspective to the paper.

**Reasons to Reject**

- Reproducibility Concerns: The paper needs to include essential implementation details of the synthetic environment. Furthermore, there needs to be more clarity about the release of necessary prompts and the step instructions' design, making it challenging for others to replicate the results and potentially question the study's validity.
- Limited Scope: The research predominantly concentrates on ChatGPT, neglecting a broader examination of other chat models. This limited focus restricts the paper's applicability and generalizability in the broader context of chatbot situational understanding.
- Evaluation Metrics Ambiguity: The difference between specific evaluation metrics, specifically Step-EM and State-EM, must be articulated, leading to potential misunderstandings about the study's conclusions.